# Overcoming Chemotherapy Resistance in Metastatic Cancer: A Comprehensive Review

**DOI:** 10.3390/biomedicines12010183

**Published:** 2024-01-15

**Authors:** Maryam Eslami, Omid Memarsadeghi, Ali Davarpanah, Afshin Arti, Karim Nayernia, Babak Behnam

**Affiliations:** 1Applied Biotechnology Research Center, Tehran Medical Sciences, Islamic Azad University, Tehran 1949635881, Iran; drmaryam.eslami2020@gmail.com (M.E.); omemarsadeghi@gmail.com (O.M.); davarpanah.alii94@gmail.com (A.D.); 2International Faculty, Tehran Medical Sciences, Islamic Azad University, Tehran 1949635881, Iran; 3Department of Genetics, Faculty of Advanced Science and Technology, Tehran Medical Sciences, Islamic Azad University, Tehran 1949635881, Iran; 4Department of Biomedical Engineering, Central Tehran Branch, Islamic Azad University, Tehran 1469669191, Iran; afshinarti@yahoo.com; 5International Center for Personalized Medicine (P7Medicine), 40235 Dusseldorf, Germany; 6Department of Regulatory Affairs, Amarex Clinical Research, NSF International, Germantown, MD 20874, USA

**Keywords:** chemotherapy resistance, metastatic cancer cells, combination therapy, targeted therapy, immunotherapy, drug repurposing, personalized medicine, predictive biomarkers, clinical trials

## Abstract

The management of metastatic cancer is complicated by chemotherapy resistance. This manuscript provides a comprehensive academic review of strategies to overcome chemotherapy resistance in metastatic cancer. The manuscript presents background information on chemotherapy resistance in metastatic cancer cells, highlighting its clinical significance and the current challenges associated with using chemotherapy to treat metastatic cancer. The manuscript delves into the molecular mechanisms underlying chemotherapy resistance in subsequent sections. It discusses the genetic alterations, mutations, and epigenetic modifications that contribute to the development of resistance. Additionally, the role of altered drug metabolism and efflux mechanisms, as well as the activation of survival pathways and evasion of cell death, are explored in detail. The strategies to overcome chemotherapy resistance are thoroughly examined, covering various approaches that have shown promise. These include combination therapy approaches, targeted therapies, immunotherapeutic strategies, and the repurposing of existing drugs. Each strategy is discussed in terms of its rationale and potential effectiveness. Strategies for early detection and monitoring of chemotherapy drug resistance, rational drug design vis-a-vis personalized medicine approaches, the role of predictive biomarkers in guiding treatment decisions, and the importance of lifestyle modifications and supportive therapies in improving treatment outcomes are discussed. Lastly, the manuscript outlines the clinical implications of the discussed strategies. It provides insights into ongoing clinical trials and emerging therapies that address chemotherapy resistance in metastatic cancer cells. The manuscript also explores the challenges and opportunities in translating laboratory findings into clinical practice and identifies potential future directions and novel therapeutic avenues. This comprehensive review provides a detailed analysis of strategies to overcome chemotherapy resistance in metastatic cancer. It emphasizes the importance of understanding the molecular mechanisms underlying resistance and presents a range of approaches for addressing this critical issue in treating metastatic cancer.

## 1. Introduction

Cancer is a leading cause of morbidity and mortality worldwide [1]. According to the World Health Organization Global Info base reports, cancer cases are on the rise, likely leading to around 15 million deaths annually. Based on the data from the American Cancer Society (ACS), in 2023, a total of 1,958,310 new cancer cases and 609,820 cancer deaths have been registered in the United States [2]. Chemotherapy is one of the most important tools in the management of metastatic cancer [3]. Chemotherapeutic drugs affect rapidly proliferating cancer cells but, unfortunately, damage normal bystander cells and also select drug resistance [4]. Multidrug resistance to the current chemotherapy drugs is the main reason for chemotherapy failures. Drug resistance is generally classified into two groups, intrinsic and acquired. Tumors with inherent resistance show a resistant phenotype to chemotherapy before they encounter chemotherapeutic drugs, which are thus ineffective in the chemotherapeutic treatment of these tumors. Most tumors with acquired resistance may respond to chemotherapy but later become insensitive to similar drugs [5]. Intrinsic and acquired multidrug resistance can limit the effectiveness of chemotherapy and have emerged as significant challenges in cancer treatment. For example, drug resistance accounts for approximately 90% of deaths in ovarian cancer patients [6]. In addition to resistance to established chemotherapeutic drugs, such as cytotoxic agents, resistance to newer-generation drugs is commonly observed in cancer cells. Therefore, the drug resistance mechanisms of cancer cells have been comprehensively investigated and novel therapeutic strategies against resistance should be further developed [6]. This review summarizes the mechanisms of drug resistance and the effect of natural products is briefly discussed. We propose a strategy to overcome drug resistance in metastatic cancerous cells that target nonapoptotic cell death, especially necroptosis, autophagy, and necrosis.

## 2. Molecular Mechanisms of Chemotherapy Resistance in Metastatic Cancer Cells

### 2.1. Genetic Alterations and Mutations Contributing to Chemotherapy Resistance

Epigenetic signatures on the DNA, histones, and other steps of transcription and translation in cancer cells with anticancer drug resistance can be readily identified. For example, certain tumor suppressor gene silencing has distinct DNA hypermethylation and might be a significant marker of carcinogenesis [7,8]. During carcinogenesis, the epigenome undergoes multiple modifications: genome-wide hypomethylation; isolated hypermethylation (expressly in CpG promoter islands of tumor suppressor genes), generalized changes in histone acetylation, and distinct microRNA (miRNA) expression. For example, demethylation of the *ABCB1* gene, which codes for a transporter protein, cancer cell lines leads to diminished intracellular accumulation of chemotherapeutic agents and leads to a multidrug resistance (MDR) phenotype. Furthermore, epigenetic alterations can affect the DNA repair system, like hypermethylation of the mismatch repair protein known as the human *MutL* homolog1 (*hMLH1*) gene, which can lead to the spread of colorectal cancer and chemotherapy resistance [9,10].

Currently, the US FDA has approved two classes of drugs that alter the epigenome:DNA methylation inhibitors (DNMTi), including 5-azacitidine and 5-aza-2′-deoxycytidine (Decitabine; DAC; histone deacetylase inhibitors (iHDACs), such as Vorinostat, Belinostat, Romidepsin, and Panobinostat. Data have shown that combining conventional chemotherapeutics with epigenetic drugs, such as DAC, can overcome chemotherapy-resistant tumors. Even though DAC does not directly affect tumor growth, it inhibits DNA methylation, which sensitizes the tumor to other chemotherapeutics, including carboplatin, cisplatin, and 5-FU [11];Colorectal cancers have distinct epigenetics. While DNA methylation in *MDF1*, *SSTR2*, *CMTM3*, *TGFB2*, and *NDRG4* genes is a potential marker for the detection of colorectal cancer in the early stages of its development, hypermethylation in the *CLDN11* gene is associated with a metastasis characteristic and a poorer prognosis [12]. Silencing of tumor suppressor candidate 3 (*TUSC3*) mRNA expression by promoter hypermethylation induces upregulation of the epidermal growth factor receptor (EGDR), leading tumor cell resistance to apoptosis [12]. DNA methyltransferase inhibitors and drugs targeting histone deacetylases could potentially be a novel anticancer strategy in this model [13]. The latest data have demonstrated that CUDC-101 and CUDC-907, newly synthesized histone deacetylase/kinase inhibitors, showed therapeutic potential as anticancer agents in colon cancer [14,15].

MicroRNAs are made up of 19–25 nucleotides and cannot code any proteins; however, they influence gene expression by posttranscriptional modifications (PTMs). Epigenetic changes associated with miRNAs frequently play an essential role in developing the chemoresistance of various types of cancer. Numerous studies have demonstrated the role of miRNAs influencing drug-resistance-related genes, genes related to cell proliferation, cell cycling, and tumor cell apoptosis; miRNAs could serve as a potential biomarker for the prognosis of the effectiveness of chemotherapy treatments [10]. The list of miRNAs involved in tumor transformation keeps on growing [16,17].

### 2.2. Epigenetic Modifications and Their Impact on Drug Resistance

Histone PTMs are proteins with specialized structures, such as chromo-(methylation), bromo-(acetylation), Breast Cancer C-terminal Domains or BCRT-(phosphorylation), and PHD domains (methylation) [18]. These proteins, such as SWI/SNF, ISWI, CHD, and INO80 chromatin remodeling, change chromatin structure and add an extra layer of regulation. The combination of DNA and histone modifications provides an enormous non-genetic diversity, allowing specificity for various biological processes. Recent efforts in mapping genetic alterations by whole genome or exome sequencing have shown that DNA- and histone-modifying enzymes and nucleosome remodelers are frequently mutated and contribute to tumorigenesis [17,18]. Establishing tumor epigenomes initiates tumorigenesis and allows the procurement of additional genetic and epigenetic changes that deregulate many biological processes, including those favoring survival in the presence of a particular drug [19]. Mechanisms involved in acquired drug resistance include increased drug efflux, inactivation of pro-apoptotic genes, perturbed DNA repair, activation of parallel or downstream signal transduction pathways, and secondary mutations in drug targets. Reversible non-genetic tumor heterogeneity is a source of drug resistance, as tumors consist of heterogeneous cell populations with different drug sensitivities. Mechanisms implicated in non-genetic drug resistance include epigenetic changes resulting in the gene transcription of drug transporters, pro-apoptotic genes, DNA-repair proteins, and histone modifiers. Withdrawal of drug treatment resulting in a drug-free period may restore susceptibility for primary cytotoxic/targeted therapy by the reversal of drug-resistance-associated epigenetic marks. Without drug-selective pressure, non-genetic drug-resistant tumor cell populations can be treated with cytotoxic or targeted drugs combined with epigenetic drugs, like HDACi, DNMTi, EPZ004777, and BET-I [18,19]. This reverses the drug-resistant epigenome into a drug-sensitive one, making tumor cells sensitive to the targeted drug. However, hard-wired genetically resistant tumor cells may still arise. A regimen combining cytotoxic and epigenetic drugs can debulk the tumor, kill the majority, and avoid genetic drug resistance toward targeted therapy [19].

### 2.3. Altered Drug Metabolism and Efflux Mechanisms

Chemotherapy resistance occurs due to a number of mechanisms, including the increased tumor cell efflux of the drug, decreased influx of the drug, drug inactivation, drug target alterations, and apoptosis evasion. The mechanism by which chemotherapy drugs are taken up by tumor cells, i.e., influx and efflux, is beginning to be elucidated [20,21]. For example, the reduced folate carrier (RFC) is used for the uptake of the dihydrofolate reductase (DHFR) inhibitor methotrexate (MTX) and the thymidylate synthase (TS) inhibitor complex. MTX resistance has been documented in tumors with RFC inactivating mutations [22].

The RFC uptake of MTX may play a key role in the efficacy of this agent. Childhood chronic myeloid leukemia (CML) with the RFC genotype (80AA) has higher plasma MTX levels, indicating decreased drug uptake and higher mortality [23].

### 2.4. Activation of Survival Pathways and Evasion of Cell Death

Tumor resistance can be due to evasion of apoptosis [24]. Caspases, by cleaving cytoplasmic or nuclear substrates, play a pivotal role in apoptosis. Caspases can initiate the secondary effector pathway [25]. The Bcl-2 family of proteins is another element in apoptosis, having both anti-apoptotic and pro-apoptotic elements. In cancer, anti-apoptotic elements are often upregulated, allowing cancer cells to evade apoptosis secondary to chemotherapy. Dysfunctional anti-apoptotic elements can also be upregulated, allowing cancer cell survival [26]. Signaling to apoptosis can be suppressed or abolished by an increase in anti-apoptotic molecules or decreased dysfunction of pro-apoptotic proteins in cancer cells [24,27]. Tumor resistance can be due to evasion of apoptosis [23]. Caspases, by cleaving cytoplasmic or nuclear substrates, play a pivotal role in apoptosis. Caspases can initiate the secondary effector pathway [24]. The Bcl-2 family of proteins is another element in apoptosis, having both anti-apoptotic and pro-apoptotic elements. In cancer, anti-apoptotic elements are often upregulated, allowing cancer cells to evade apoptosis secondary to chemotherapy. Dysfunctional chemotherapy-induced apoptosis is important in drug resistance. Overexpression of anti-apoptotic proteins, such as Bcl-2, can be detected in many cancer cells; this can result in a lack of responsiveness to chemotherapy. Dysfunctional anti-apoptotic elements can also be upregulated, allowing cancer cell survival [25]. Signaling to apoptosis can be suppressed or abolished by an increase in anti-apoptotic molecules or decreased dysfunction of pro-apoptotic proteins in cancer cells [24,26].

Pro-apoptotic BH3-only proteins, such as Bim, Bid, BAD, and Noxa, promote apoptosis and act as tumor suppressors. Defects in the mitochondrial pathway of apoptosis may also occur at the post-mitochondrial level [28,29]. The translationally controlled tumor protein (TCTP) protects tumor cells from apoptosis via interference with the Apaf-1 complex, preventing the caspase cascade from happening. Overexpression of the TCTP often leads to chemotherapy resistance by dysregulating the formation of apoptosis. One possible reason for the resistance of cancer cells to apoptosis may be the dysfunction of inhibitors of apoptosis proteins (IAPs) in humans, which are a family of endogenous caspase inhibitors with eight members. XIAP shows the most potent anti-apoptotic characteristics. Furthermore, XIAP prevents apoptosis by preventing caspase-9 activation.

IAP expression and function are dysregulated in human cancers in which IAPs and NF-κB play a pivotal role in evading apoptosis, with chemotherapy resistance with poor prognosis in acute myeloid leukemia. X-linked lymphoproliferative syndrome type 2 is caused by a deficiency in XIAP. Small-molecule inhibitors of IAPs might be beneficial in inducing cancer apoptosis. Resistance to Fas-mediated apoptosis leads to chemotherapy resistance and poorer prognosis. Nucleolin, an activation-resistant Fas complex, inhibits Fas-mediated apoptosis in B-cell lymphomas. The down-regulated expression of CD95 has been observed in drug-resistant leukemia and neuroblastoma cells [30,31].

Alterations of the CD95 gene were also uncovered in solid tumors and hematological malignancies [31]. Deficient transport of the apoptosis-inducing tumor-necrosis-factor-related apoptosis-inducing ligand (TRAIL) receptors TRAIL-R1 and TRAIL-R2 from the endoplasmic reticulum to the cell surface presented opposition to TRAIL-induced cell death in colon cancer cells [32]. Loss of function of the apoptosis-inducing TRAIL receptors has been reported in numerous cancers [32]. Changes in any of the decoy receptors are another mechanism of evading TRAIL- or CD95-induced apoptosis. Gastric cancers have been shown to have overexpression of TRAIL-R3, a decoy receptor for TRAIL. The signal transduction through the death receptor passageway can be functionally modified at the receptor level. Cellular FLICE (FADD-like IL-1β-converting enzyme)-inhibitory protein (c-FLIP), a major anti-apoptotic regulator and resistance factor, can repress the recruitment of caspase-8 and, thus, suppress TRAIL-mediated apoptosis, triggered by chemotherapeutic agents in cancers.

Several signaling molecules can modulate apoptosis and its signaling pathways and, thus, change the sensitivity of anticancer cells to chemotherapy [30,31,32]. For example, several Bcl-2 family proteins, pro- and anti-apoptotic family proteins, are controlled by Jun amino-terminal kinase (JNK) and p38-MAPK. JNK can effectively promote the p53 upregulated modulator of apoptosis (PUMA) activation and apoptosis in chemo-resistant cells through the Akt/Fox03a pathway. Insulin-like growth factor 1 suppressed 5-fluorouracil-induced apoptosis via both the phosphatidylinositol 3-kinase (PI3K)/Akt and casein kinase pathways, which arrested Smac/DIABLO release and blocked the activation of caspases [33]. Mutations in p53 caused chemoresistance through its possessions on mitochondrial works and restored p53 function, leading to chemosensitivity. Overexpression of metalloproteinase-1 tissue inhibitors linked to a compromising response to chemotherapy via phosphatidylinositol 3-kinase PI3K/Akt and the NF-κB signaling pathway [33,34]. Furthermore, the actin-bundling protein (fascin) acts as a pivotal mechanism of breast cancer chemoresistance; this is conducted via the enhanced expression of anti-apoptotic proteins and suppression of the pro-apoptotic proteins, like caspase-3 and -9 [33,34].

## 3. Strategies to Overcome Chemotherapy Resistance in Metastatic Cancer Cells

### 3.1. Combination Therapy Approaches and Rationale behind Their Effectiveness

Nanoparticles loaded with both small-molecule MDR modulators and chemotherapeutic drugs are known as combinatory nanoparticles. They have shown promise in restoring drug resistance. Curcumin has been widely studied because it is an effective MDR modulator and exhibits certain anticancer properties. Ganta et al. reported on the coadministration of curcumin and paclitaxel using nano-emulsions in human ovarian cancer cell lines [35]. The curcumin down-regulates P-gp expression and inhibits NF-κβ activity, which leads to the increased uptake of paclitaxel and enhanced apoptosis within multidrug-resistant (MDR-1 positive) SKOV-3TR cells. In another study, Amornwachirabodee et al. modified curcumin molecules with hy-653 hydrophilic methoxy polyethylene oxide (PEO) and aquaphobic palmitate (PA) [36]. These amphiphilic curcumin molecules (mPEO-Cur-PA) automatically self-assembled into bilayer vesicular spheres in aqueous circumstances and showed enhanced cellular uptake via endocytosis. Pramanik et al. formulated curcumin in doxorubicin (DOX)-conjugated polymeric nanoparticles to overcome DOX resistance in human and murine cancer cell lines [37]. Curcumin strongly suppressed the MDR phenotype in DOX-resistant cancer cells. These nanoparticles exhibited significant inhibition of subcutaneous tumor growth (N90%) in DOX-resistant human melanoma (RPMI8226 xenograft) models while cardiotoxicity in mice was substantially reduced. Similarly, Misra et al. reported the down-regulation of P-gp expression and the enhanced cytotoxic effect of DOX upon the coadministration of DOX and curcumin using poly(D, L-lactide-co-glycolide) nanoparticles [38]. Curcumin not only enhanced the nuclear concentration of DOX but also inhibited *MDR1* and *Bcl-2* at the mRNA level in the human leukemia cell line. Reversal of drug resistance in NCI/680 ADR-RES cell lines (referred to as MCF-7/ADR cell lines in this paper) upon coadministration of DOX and curcumin using chitosan/poly(butyl cyanoacrylate) nanoparticles has also been reported [39]. It is worth mentioning that multidrug-resistant MCF-7/AdrR cell lines were thought to be derived from MCF-7 breast adenocarcinoma cell lines; however, some recent studies found that MCF-7/AdrR cell outlines were essentially derived from OVCAR-8 human ovarian carcinoma cells [40]. Ye et al. reported the effect of curcumin on the reversal of cis-platin (cis-diamminedichloroplatinum (II); DDP) resistance in cis-platin-resistant human lung adenocarcinoma cell lines (A549/DDP). Hypoxia-inducible factor (HIF)-1α also contributes to cisplatin resistance in A549/DDP cells under normal conditions. Curcumin interrupts HIF-1α expression at protein levels, promotes HIF-1α degradation, and reverses cisplatin resistance in A549/DDP cells. Expression of P-gp, a downstream target of HIF-1α, also decreased in response to curcumin in a dose-dependent manner. Furthermore, curcumin triggered the apoptosis of A549/DDP cell lines via a caspase-3-dependent mechanism. Therefore, such nanoparticles offer powerful and versatile strategies for treating multidrug-resistant cancers [41,42,43,44]. Recent studies have demonstrated that the coadministration of antitumor drugs and siRNAs offers another strategy to enhance the therapeutic efficacy of cancer treatment in various multidrug-resistant cancer models [45]. SiRNAs are known to down-regulate the expression of multidrug-resistant proteins and sensitize the cancer cell for the cytotoxic action of anticancer drugs. Navarro et al. reported P-gp silencing with siRNA delivered using dioleoylphosphatidylethanolamine-modified polyethyleneimine (DOPE-modified PEI) nanoparticles in NCI/ADR-RES [46]. The down-regulation of P-gp managed to inhibit DOX efflux, leading to increased intracellular DOX accumulation and enhanced drug toxicity in DOX-resistant human breast cancer cell lines. Increased expression of the signal transducer and activator of transcript (Stat3) proteins leads to chemotherapy resistance. Su et al. used poly lactic-co-glycolic acid (PLGA) nanoparticles for the co-delivery of Stat3 siRNAs and paclitaxel [47]. Paclitaxel was loaded into the PLGA nanoparticles through a solvent evaporation method. Next, the nanoparticles were coated with a cationic PEI layer, enabling them to carry Stat3 siRNAs on their surface through electrostatic binding. These nanoparticles suppressed Stat3 expression and induced apoptosis in human lung cancer cell lines (A549) and A549-derived paclitaxel-resistant A549/T12 cell lines with an α-tubulin mutation. In another study, poly [bis-(2hydroxyethyl)-disulfide-diacrylate-β-tetraethylenepentamine], a poly(β-amino esters)-based nanoparticle system, was used for the co-delivery of MDR-1-shRNA, iSurvivin-shRNA, and DOX into a mice xenograft model with NCI/ADR-RES tumors [48]. These complexes down-regulated the expression of P-gp and Survivin. They lowered the IC50 value of DOX in MDR cancer cells by 6.4 to 766.3 folds, attributed to an increase in the intracellular DOX accumulation, cell cycle arrest, and recovery of the blocked cell apoptosis pathway. Recently, Hao et al. demonstrated the reversal of MDR by RNAi in a human kidney carcinoma cell line (RCC A498) [49]. Stable RCC A498-RNAi cells (RCC A498 cells transfected by shRNA recombinant plasmid) and control cells (RCC A498 cells transfected by random RNA recombinant plasmid) were prepared. The downregulation of MDR1 mRNA and P-gp expression reduces the IC50 of the anticancer drugs in the RCC A498-RNAi cells. The use of silica nanoparticles for the co-delivery of DOX and Bcl-2 siRNA was explored by Chen et al. [50]. Polyamidoamine dendrimers (PAMAM) were conjugated onto mesoporous silica nanoparticle surfaces to introduce positive surface charges. Additionally, siRNAs targeted against mRNA encoding Bcl-2 protein were effectively complexed onto these nanoparticles. The anticancer efficacy of DOX increased drastically (~132 times) in multidrug-resistant human ovarian cancer cell lines (A2780/AD) because the co-administrated siRNAs significantly suppressed Bcl-2 mRNA. Several research groups also studied the coadministration of curcumin and siRNAs to enhance the effectiveness of chemotherapy in drug-resistant cancer models. The Notch signaling path plays an important role in human cancers, such as osteosarcoma, suggesting that discovering specific agents that target the Notch pathway would be efficacious for osteosarcoma therapy. Dhule et al. demonstrated the curcumin-triggered inhibition of osteosarcoma cell growth and G2/M phase cell cycle arrest [51]. Down-regulation of *Notch-1* via siRNA before curcumin treatment also enhanced cell growth but inhibited invasion. Although both curcumin (10 μM) and Ki-67-7 siRNAs (10 nM) showed a moderate (~60%) inhibitory effect on the cell viability of the AY-27 and T-24 cell lines, coadministration of curcumin and siRNAs increased cell viability to over 80%. They also demonstrated that the rate of apoptosis of AY-27 cells treated with both curcumin and Ki-67-7 (36%) was greater than that of those treated with Ki-67-7 siRNAs (14%) or curcumin (13%) alone. The transcription factor Wilm’s tumor gene 1 (*WT1*) is crucial in cancer, including pancreatic cancer. Glienke et al. studied the effect of curcumin and siRNAs working against *WT1* on its expression using the pancreatic cancer cell line PANC-1 [52]. The *WT1* mRNA levels were diminished by 20 and 90%, respectively, in response to 10 μM curcumin and 2.5 μg *WT1* siRNA. The combination of *WT1* siRNA (2.5 μg) and curcumin (10 μM) resulted in a marginal decrease in *WT1* mRNA compared to WT1 siRNA alone. However, combined treatment with curcumin (40 μM) and *WT1* siRNA 787 (2.5 μg) inhibited cell proliferation more effectively (~80%) than treatment with curcumin alone (~60%) [52].

### 3.2. Targeted Therapies and Their Potential in Overcoming Drug Resistance

Nanoparticles have a significant benefit as anticancer agents [53]. The speedy growth of solid tumors results in altered physiology at the tumor site, which leads to leaky vasculature. The increased vascular permeability and decreased lymphatic drainage in the tumor produce an enhanced permeability and retention effect (EPR effect) [54]. Physicochemical factors, such as magnitude and distribution, surface charge, and surface hydrophobicity, play a main position in inactive targeting. Subdivisions less than 100 nm can pass through the liver endothelium’s fenestrations and the sine curve’s separate plates to localize in the liver. Nanoparticles also tend to accumulate in the reticuloendothelial system (RES), the spleen, and bone marrow [55]. PEGylating can enhance the solubility and stability of nanocarriers in an aqueous environment while minimizing opsonization during circulation. This will substantially boost the nanocarriers’ circulation time, thereby enhancing the drug nanocarriers in tumors [56,57]. Several other biocompatible and hydrophilic polymers with flexible main chains have been explored as hydrophilic shells for nanocarriers, including poly(acrylamide), poly(vinylpyrrolidone), and poly(vinyl alcohol) [58]. The extent of passive tumor targeting is limited. Thus, tremendous efforts have been directed toward developing active tumor targeting by conjugating nanocarrier systems with ligands restricted to receptors overexpressed in cancer cells compared to normal cells. Various biological molecules, including peptides, antibodies, aptamers, and other small molecules, including folic acid, have been used to target tumor cells [59]. More importantly, nanoparticles can bypass drug efflux by ABC transporters since they are adopted via either non-specific or specific endocytosis [58]. There are four leading systems of endocytosis: clathrin-mediated endocytosis, caveolae-mediated endocytosis, macropinocytosis, and clathrin- and caveolae-independent endocytosis. Among them, clathrin-mediated endocytosis is the most widely studied mechanism for receptor-mediated uptake of drug nanocarriers. Some of the well-known receptors associated with this mechanism of endocytosis are transferrin, low-density lipoprotein (LDL), epidermal growth factor receptors (EGFRs), human epidermal growth factor receptor 2 (HER2), and lectins [58]. For clathrin-mediated endocytosis, soluble clathrins assemble into clathrin triskelia in the cytoplasm, recruited at ligand-receptor binding sites on the plasma membrane. These clathrin triskelia multimerize and form a polyhedral lattice, which helps to deform the membrane into a coated pit that buds and pinches off from the film in a dynamin-dependent manner forming clathrin-coated vesicles. These clathrin-coated vesicles, which are uncoated after endocytosis, fuse with the l endosome and, eventually, lysosomes. The acidic environment of lysosomes stimulates and catalyzes the degradation of the internalized drug nanocarriers to release their payloads. Caveole-mediated endocytosis is likely the most common clathrin-independent receptor-mediated endocytosis [58]. Caveolae are flask-shaped membrane invaginations formed by oligomerized caveolins and are present in many cell types, such as smooth muscle cells and adipocytes [60]. For caveolae-mediated endocytosis, after fusing with the cell membrane, nanocarriers move along the membrane to the caveolae invaginations described as a lipid raft, which subsequently detach from the cell membrane via a dynamin-mediated process and generate cytosolic caveolar vesicles. These caveolar vesicles can fuse with early endosomes or caveosomes where they do not undergo acidification. Ligands specific to caveolae-mediated endocytosis include folic acid, albumin, and cholesterol [61]. Macropinocytosis is a non-selective endocytosis process in which actin-driven membrane protrusions filled with extracellular fluid containing soluble and dispersed materials, such as nanocarriers, are pinched from the cell membrane, forming macropinosomes. The macropinosomes may progress onto the endolysosomal system or fuse back with the cell membrane [58,61]. Various clathrin- and caveolae-independent endocytosis pathways, such as flotillin-dependent endocytosis and RhoA (Ras homolog gene family, member A)-dependent endocytosis, are also known. Many of the chemotherapeutic agents (nearly 40%) or MDR modulators are highly hydrophobic, which leads to poor aqueous solubility and low bioavailability. For example, the aqueous solubility of curcumin is less than 20 μg/mL [58]. Moreover, at physiological pH, the soluble fraction of curcumin undergoes rapid hydrolysis, followed by molecular fragmentation [58]. Wang et al. demonstrated that curcumin decomposed (N90%) within a half-hour of incubation at 37 °C in 0.1 M phosphate manila and serum-free medium [62]. These undesirable pharmacological properties of free anticancer drugs/MDR modulators can be improved using nanocarriers because nanocarriers can enhance the solubility of hydrophobic drugs in aqueous solutions and provide sustained and controlled drug release. Furthermore, it enhances solubility and prevents drugs from premature in vivo degradation. Dhule et al. demonstrated the enhanced aqueous solubility of curcumin (~600 μg/mL) in a 2-hydroxypropyl γ-cyclodextrin/curcumin-liposome complex [51]. Solid lipid nanoparticles (SLNs) loaded with curcuminoids for topical use have also been developed. The light and oxygen sensitivity of curcuminoids can be overcome by incorporating curcuminoids into this unique type of formulation [51].

### 3.3. Immunotherapeutic Strategies for Enhancing the Immune Response against Metastasis

Cancer is a complex disease characterized by dysregulated growth and expansion [63]. In 1983, William Coley used live bacteria as an adjuvant to target cancer successfully, highlighting the importance of the immune systems relevant to cancer recognition and clearance. Furthermore, the immune system plays an important role in cancer [64]. Tumors can escape immune surveillance through various strategies, such as the secretion of cytokines like vascular endothelial growth factor (VEGF) and Fas-L expression. Antigen presentation is adversely affected by the downregulation of the major histocompatibility complex (MHC) [65]. Nanocarriers can further enhance cancer immunotherapy by targeting drugs to T-cells [66,67,68].

Recent developments in novel immune-based approaches to fight cancer include the cancer immunosurveillance theory, which focuses on the interaction of cancer cells with the immune system. Various factors, including altered antigenicity and soluble factors, can cause antitumor immunosuppression. NK cells detect transformed cells and engulf them, presenting tumor-derived molecules to B- and T-cells and secreting inflammatory cytokines [69]. B- and T-cell activation induces the secretion of cytokines, which boost innate immunity and promote the production and expansion of antibodies and tumor-specific T-cells [70,71]. The adaptive immune system produces immunological memory and prevents tumor recurrence therapy [72].

Cytokine therapy is one such therapy that can increase antitumor responses. Cytokines in the tumor microenvironment (TME) play a critical role in cancer pathogenesis. Targeting cytokines in TME is an effective approach in cancer [65,72]. Dendritic cell therapy has been proven to be a well-tolerated, efficacious, and safe immunotherapeutic strategy, eliciting antitumor immunity, even in advanced-stage cancers [66,67]. Adoptive T-cell therapy (ACT) involves cytotoxic T-cells recognizing, targeting, and destroying tumor cells [71]. ACT has induced regression in melanoma p and can harness a CD4^+^ T-cell response against a mutated cancer antigen erbb2-interacting protein for metastatic epithelial cancer regression. Chimeric antigen receptor (CAR) T-cell therapy has shown meaningful clinical efficacy in melanoma and non-small-cell lung cancer patients [69]. CAR-based therapy targets antigens on cells destroyed by reconstructing the CAR, which may involve the viral vector-based retrotransposon [69]. CAR T-cell therapies are classified into four distinct types based on their cytokine profile and loss of co-stimulatory molecules. The US FDA approved two CAR T-cell therapies in 2017 for acute lymphoblastic leukemia and advanced lymphomas in adults [72]. Three sources for tumor-specific T-cells elicit adoptive immunotherapy are employed: (1) tumor-infiltrating lymphocytes, (2) genetic engineering of T-cell populations, and (3) autologous lymphocyte-activated killer cells. Immunotherapy using autologous NK cells has shown promising results [73].

Neoantigens can be used to develop novel therapeutic approaches that selectively enhance specific T-cell reactivity. NK cells can also play a significant role in triggering the body’s immune response against cancer [69]. There are several NK cell-based cancer immunotherapeutic systems to overcome NK cell paralysis. These include stable allogeneic NK cell lines, expanded NK cells uninhibited by self-histocompatibility antigens, genetic modification of NK cell lines, and fresh NK cells for prolonged expression of Fc receptors or chimeric tumor-antigen receptors and cytokines [69,72]. Immune checkpoint inhibitors, such as cytotoxic T-lymphocyte-associated antigen-4 antibodies, can block or interrupt immune checkpoints, unleashing antitumor immunity. These pathways are crucial for maintaining self-tolerance, amplitude, and duration of physiological immune responses. Blockers of additional immune-checkpoint proteins, such as programmed cell death protein-1, have been shown to produce effective clinical outcomes. Gc protein-derived macrophage-activating factor (GcMAF) is a highly polymorphic serum protein synthesized by the liver and has been used as an effective immunomodulator in cancer treatment [74]. Antibodies have emerged as efficacious therapeutic agents due to their immunomodulatory properties. Therapeutic monoclonal antibodies targeting various types of cancers have been developed. Various therapeutic vaccines are currently being developed under clinical trials including HBV and HPV vaccines for liver and cervical cancer and immune cell-based vaccines like Sipuleucel-T [75,76]. Nanomaterial-based immunotherapy has enhanced immunotherapy, photoacoustic imaging, and photothermal therapy in treating gastric cancer. These advancements have solved traditional immunotherapy problems and can potentially improve clinical outcomes and enhance antitumor immunity [73].

### 3.4. Repurposing Existing Drugs and the Identification of New Therapeutic Targets

In comparison to ground-up drug development, repurposing known and approved drugs for cancer is a promising route [76] and, hence, has sparked a growing concern in drug repurposing [77]. This unmet need for more successful anti-cancer drugs has sparked a growing concern in drug repurposing [77].

Cardiac glycosides are used to treat various heart conditions. They inhibit the Na^+^/K^+^-ATPase ion pump and increase intracellular sodium levels [77]. These drugs have anti-proliferative effects on HeLa cells and other cancer cells. They also induce the expression of the cyclin-dependent kinase p21Cip1, leading to growth arrest [77]. This class of drugs can eradicate recognized xenograft cancers through suppressing HIF-1α and HIF-2α. Aspirin works by blocking cyclooxygenase (COX) enzymes 1 and 2 [77,78]. Aspirin is associated with favorable survival in patients with colorectal cancer; however, most studies are retrospective and heterogeneous. Future prospective studies are needed to identify the role of aspirin as a therapeutic chemotherapy agent.

COX1 produces thromboxane A2 in platelets, leading to platelet aggregation and adherence to tumor cells [79,80,81]. COX2 is responsible for producing prostaglandin E2, which promotes proliferation. Aspirin has been studied for its therapeutic potential in established or chemically induced tumors. However, most studies are retrospective and heterogeneous, with conflicting findings. The role of aspirin as a therapeutic cancer agent was shown previously [82,83,84].

Beta-blockers target beta-adrenoreceptors, which are activated by catecholamines and promote tumorigenesis. Blocking beta-blockers can induce anti-proliferative and anti-migratory effects and increase overall survival in experimental animal models of pancreas and colorectal cancer [84,85].

Metformin is used to treat diabetes type II; it inhibits oxidative phosphorylation, energetic stress, and gluconeogenesis. Its anti-neoplastic effects depend on cancer cells’ ability to activate glycolysis-mediated ATP production and glucose availability [86,87]. Metformin indirectly activates adenosine monophosphate-activated protein kinase (AMPK) through the tumor suppressor liver kinase B1 (LKB1) [88]. AMPK inhibits the unfolded protein response (UPR) in leukemic cells, resulting in tumor cell death [89]. Metformin medicine may cause the growth of ER stress, leading to apoptosis [88]. At standard diabetic doses, it can reduce the number of proliferating cells. It has been inferred that tumors respond to metformin depending on their molecular subtype [89].

Chlorpromazine (CPZ), a derivative of phenothiazine, has been found to inhibit tumor growth in various cancers, including hepatocellular carcinoma, glioma, leukemia, and melanoma [90], most likely through the altered expression of cell cycle-related proteins and interference with mitochondrial processes. Chlorpromazine depresses mitochondrial ATP production and DNA polymerase activity in leukemic cells [91]. CPZ induces autophagy by inhibiting the Akt/mTOR pathway, as shown in glioma cells. Penfluridol, a first-generation antipsychotic drug, exerts anti-proliferative effects through interference with integrin signaling. Fluspirilene, an anti-psychotic drug, has been identified as a cancer drug candidate due to its ability to bind to murine double minute 2 (MDM2), potentially blocking the interaction between p53 and MDM2 [92].

Tricyclic antidepressants have antineoplastic effects by inhibiting mitochondrial complex III, leading to caspase activation and apoptosis [93].

Amitriptyline, a tricyclic antidepressant, is suggested for use as an oxidation therapy agent [94].

Lithium (LiCl) has traditionally been used for bipolar disease, mainly due to its inhibition of glycogen synthase kinase 3, which impacts multiple cellular functions [95,96]. LiCl has shown growth-inhibitory effects in prostate cancer cell lines and tumor xenografts through GSK3 inhibition. It has also been shown to increase the effect of doxorubicin and etoposide in prostate cancer cell lines [96]. LiCl can prevent metastasis by inhibiting lymphangiogenesis, down-regulating Smad3, and transforming growth factor beta-induced protein (TGFBIp) [97]. Artesunate, a derivative of artemisinin, has anti-angiogenic effects in lymphoma and myeloma cells and hepatocellular carcinoma [98]. Dihydroartemisinin shows growth-inhibitory effects on leukemia cells through ROS-dependent autophagy and subsequent apoptosis. Mebendazole, a common drug used to treat parasitic worm infections, has antiparasitic effects by inhibiting tubulin polymerization [99]. Mebendazole (MBZ), an antitumor drug, has shown promising results in various cancer types, including lung, colon, melanoma, glioblastoma, and medulloblastoma [100,101,102]. In vitro studies showed that MBZ inhibited lung cancer cell growth, resulting in smaller tumors and reduced metastases. In melanoma, MBZ induced apoptosis through caspases and Bcl-2 while, in colon cancer, MBZ was found to have high binding affinities and potentially be an inhibitor of kinases and oncogenes [103]. Combining MBZ with temozolomide, routinely used for treating malignant gliomas, inhibited tumor growth more than temozolomide alone. MBZ has antitumor effects. Itraconazole, an anti-fungal drug, blocks 14-α-lanosterol demethylase. It has been shown to also block angiogenesis and endothelial cell proliferation [104,105].

A hedgehog inhibitor through smoothened has been proposed for cancer treatment. Itraconazole with standard chemotherapy for lung cancer patients increased both progression-free and metastasis-free survival [106].

Itraconazole has some contraindications due to potential interference with antifungal drugs, such as molecular antibodies like rituximab. Protease inhibitors (PIs) have been used in the HIV treatment known as HAART. Ritonavir, one example of a PI, can block cell cycle progression and induce apoptosis through multiple mechanisms [107].

It has been shown to increase the efficacy of temozolomide in glioma cell lines, increase the effect of the chemotherapeutic drug tretinoin (ATRA) in leukemic cells, and promote cancer cell differentiation. Ritonavir co-administered with a proteasome inhibitor bortezomib had limited efficacy in solid tumors [108]. It has also been found to be a histone deacetylase inhibitor and, thus, can be considered as an epigenetic altering drug [109].

Nelfinavir, another protease inhibitor, also has anticancer properties through the inhibition of Akt-signaling and induction of ER stress. Treatment can reduce the phosphorylation of STAT3 and Akt and decrease the expression and secretion of the VEGF in various cancers [110].

Nelfinavir is a potent inducer of endoplasmic reticulum stress and unfolded protein responses in various cancers, including non-small-cell lung, ovarian cancer, liposarcoma, and breast cancer cells. In liposarcoma cells, it results in increased levels of sterol regulatory element binding protein-1 (SREBP-1) and activating transcription factor 6 (ATF6), inhibiting the enzyme site-2-protease (S2P). In breast cancer cells, a combination of, nelfinavir/COX-2 inhibition combined with drugs inhibiting autophagy could enhance cytotoxic effectivity [109]. Nelfinavir and bortezomib, augment ER stress and growth inhibition in vitro and in vivo [111,112].

A combination of nelfinavir and bortezomib has shown potential in terms of worth and toxicity profile. Tetracyclines, such as doxycycline, have been shown to inhibit angiogenesis and have growth-inhibitory effects in various cancers. Doxycycline was studied in cancer metastasis, showing potential in reducing tumor volume and soft tissue surrounding bones. It has also decreased metastasis through matrix metalloproteinase (MMP)-2/9 inhibition in prostate cancer and oral squamous cell carcinoma. Treatment with doxycycline can decrease epithelial-mesenchymal transition (EMT) markers in lung cancer and hepatocellular carcinoma cells, reversing pro-metastatic phenotypes [113].

However, a phase II clinical trial for metastatic renal cell carcinoma found no benefit and doxycycline was found to have significant systemic toxicity over time.

Nonsteroidal anti-inflammatory drugs (NSAIDs) act by repressing the formation of eicosanoids, mostly by inhibiting COX-1 and COX-2. NSAIDs are chemopreventive and can be effective in treating established tumors. They also suppress NF-kappa-B-activation and reduce cancer cell growth by inducing the gene MDA-7 (IL-24) [113,114].

Ibuprofen can inhibit prostate cancer cell growth and exhibit antitumor properties. It has been shown to have antineoplastic properties in various cancer types, including fibrosarcoma, hepatoma, colon, ovarian, and pancreatic cancer [114]. It can also inhibit cell proliferation, trigger apoptosis, and suppress metastasis. Diclofenac has shown antineoplastic effects in various tumor types, including ovarian, breast, brain, colon, pancreatic, lung, liver, and leukemic cancer. It has also demonstrated tumor inhibition in murine pancreatic cancer and ovarian cancer. It is FDA-approved as a 3% topical gel for the treatment of pre-cancerous actinic keratosis. Leflunomide, an immunomodulatory drug, has been shown to inhibit growth, disrupt cell cycle regulation, and induce apoptosis in the thyroid, neuroblastoma, chronic lymphocytic leukemia, and glioma cells. Leflunomide also modulates the chemosensitivity of several cancer cell lines and resistance to chemotherapeutic drugs has been observed in chronic lymphocytic leukemia. Fludarabine is a common cancer treatment that counteracts fludarabine’s effects by increasing CD40/IL-4 signaling and STAT signaling [115]. Leflunomide treatment decreases STAT3/6 phosphorylation and induces apoptosis by inhibiting NF-κB and anti-apoptotic proteins. It may also inhibit angiogenesis by decreasing VEGF expression and microvessel density. Auranofin, a gold complex used to treat arthritis, has anticancer properties by inducing apoptosis in various cancers [115,116]. It is a potent inhibitor of the redox enzyme thioredoxin reductase and inhibits the ubiquitin–proteasome pathway, which is upregulated in many cancers. Thalidomide, a drug derivative of glutamic acid, has shown antineoplastic effects in non-small-cell lung cancer and leukemia. Initially launched as a sedative in 1957, it was retracted in 1961 due to its teratogenic effects. Thalidomide has since been FDA-approved for erythema nodosum leprosum (ENL) and has been used for various diseases, including skin disorders, infectious diseases, immunologic disorders, and cancers [115]. It modulates signaling pathways deregulated in cancer cells, inhibiting tumor necrosis factor-α and NF-κB. Thalidomide also inhibits interleukin-1β, IL-6, IL-12, granulocyte-macrophage colony-stimulating factor, vascular endothelial growth factor, basic fibroblast growth factor, and interferon-γ [116]. However, its results in solid organ malignancies have not been as promising, with some cancers showing modest responses. Thalidomide may be a promising agent in treating hormone-dependent prostate cancer; however, toxicity issues have hindered its development [116].

## 4. Prevention of Chemotherapy Resistance in Metastatic Cancer Cells

### 4.1. Strategies for Early Detection and Monitoring of Resistance Development

Molecular resistance mechanisms can be divided into intrinsic and acquired tolerance models. Intrinsic resistance refers to the organism’s characteristics that have evolved to have resistance properties [117,118,119]. Classification of a resistance is based on whether the resistance is made or not, depending on how it occurs. Intrinsic resistance exists before drug treatment; whereas, acquired resistance is induced after treatment. Both types of resistance are seen in approximately 50% of cancer patients [118]. Intrinsic resistance is generally defined as innate resistance that occurs before drug administration. This innate resistance often leads to decreased treatment efficacy. On the other hand, acquired resistance can be recognized by a steady decrease in the anticancer effectiveness of a drug following its administration [118,119,120]. Chemotherapeutic resistance, whether it be inherent or acquired, is brought about and maintained through a decrease in drug accumulation and an augmentation in drug export, modifications in drug targets and signaling transduction molecules, an intensified mending of drug-induced DNA destruction, and the avoidance of apoptosis [121]. Also, in intrinsic resistance the presence of resistance mechanisms exists prior to the initiation of treatment. The etiology of this resistance is multifaceted and encompasses various factors, including the presence of therapy-resistant cell populations, the manifestation of low tolerance to the therapy in the patient or the occurrence of intolerable side effects, and the therapy’s inability to attain the necessary pharmacokinetic profile due to altered absorption, distribution, metabolism, or excretion. In contrast to intrinsic mechanisms, acquired resistance can be characterized by the emergence of drug-resistant cell populations harboring secondary genetic alterations that arise during treatment. Similar to intrinsic resistance, acquired resistance ultimately culminates in therapy failure. The subsequent exemplifications delineate a limited number of obtained resistance mechanisms: [117] augmentation of the rates of the extrusion of drugs or diminished rates of the influx of drugs into the neoplastic cells, which are mediated through transmembrane transporters responsible for drug uptake and extrusion; [118] biotransformation and the metabolism of drugs, predominantly induced by CYPs (Cytochromes P450s) present in the neoplasm; [119] modification of the the function of DNA repair and hindered apoptosis; [120] the impact of epigenetics/epistasis, specifically methylation, acetylation, and altered levels of microRNA, leading to modifications in the upstream or downstream effectors; [122] mutation of the drug target in targeted therapy and alterations in the cell cycle and its checkpoints; and the neoplastic microenvironment. Combining these mechanisms can potentially induce chemotherapy resistance in the cancer context [123].

For example, the effectiveness of methotrexate depends on its transport into cells by reducing folate transporter 1 (*RFT-1*), its subsequent conversion to long-lived cellular polyglutamate compounds, and the maintenance of *DHFR*. This combination effectively inhibits thymidylate and purine synthesis while promoting apoptosis. Cellular defects in one of these steps will cause the reaction. It has been established the mutations in *RFT-1*, expansion or mutation of *DHFR*, loss of polyglutamic acid, and dysfunction of the apoptotic pathway can lead to ineffective methotrexate [117,118,119,120,121]. It is expected to receive backlash. It occurs during treatment, similar to intrinsic resistance, eventually leading to treatment failure. Resistance mechanisms include, but are not limited to, increasing the rate of drug efflux or decreasing the rate of drug efflux into tumor cells; this is facilitated by the role of the transmembrane carrier function for drug uptake and efflux. Biotransformation and drug metabolism, mainly mediated by CYP (cytochrome P450) in tumors, also contribute to disease transmission. Additionally, altered DNA repair and impaired apoptosis may contribute to this attack. Epigenetic factors, such as methylation, acetylation, and changes in microRNA levels, have also been shown to play a role in developing drug resistance. The biggest challenge in cancer treatment lies in metastatic cells. Although chemotherapy is an effective treatment for cancer, outcomes and outcomes for cancer patients remain poor. Cancer cells are sensitive to almost all chemotherapy drugs by various mechanisms and methods. Although no progress has been made in reducing the incidence of new cancers, significant progress has been made in prevention and treatment strategies and a decrease in cancer and cancer cases has been achieved [122].

This achievement is mainly due to the reduction in cancer through preventive measures. To achieve these health benefits, early detection of cancer is crucial. Approximately 80–90% of cancer mortality can directly or indirectly relate to chemo-resistance. This resistance may be specific to a specific drug or involve multiple drugs with different mechanisms of action, known as MDR (multidrug resistance). Rapid drug resistance diagnosis can help predict cancer cells’ sensitivity. Early diagnosis can reduce cancer risk; however, effective screening must demonstrate the ability to detect asymptomatic tumors years earlier than conventional screening in long-term studies. The benefit of early cancer diagnosis is that patients live longer (91% for early diagnosis, 26% for late diagnosis). The main reason behind this significant success is that the tumor can be removed by surgery or treated with fewer cancer drugs, thus reducing the rate of tumor recurrence and the need to repeat or use chemotherapy [124]. Therapies used to treat cancer include surgery, chemotherapy, combined radiation therapy, and laser therapy [125]. Since metastasis accounts for more than 90% of cancer patients, addressing and managing this lifelong health problem requires maximum effort. Unlike cancer, which can often be treated with local surgery or radiation, it is a metastatic disease. Therefore, screening, chemotherapy, treatment, and immunotherapy form the basis for preventing and treating metastasis [125].

Whether intrinsic or acquired, drug resistance results from reduced intracellular drug accumulation, increased drug clearance, altered drug targets, modified signaling molecules, and maintenance of apoptosis. The development of chemoresistance is driven by decreased intracellular activation of pro-drugs (e.g., thiotepa and tegafur) or increased drug clearance. Reversal of chemoresistance can be achieved through the use of drugs and chemicals. More research is needed to understand how cancer cells respond to anti-cancer drugs and identify new strategies to overcome this resistance [117,118,119,120,121,122,123,124,125]. Resistance of cancer cells to anti-cancer drugs can be attributed to many factors, such as genetic changes in somatic cells in the tumor. In addition, anti-cancer drugs spread and many drugs may result from multiple mechanisms, such as the inhibition of cell death, changes in drug metabolism, epigenetic modification, altered drug targets, and DNA repair [126]. Cancer treatment has faced many challenges, including but not limited to resistance to cytotoxic agents and toxicity of chemotherapy. To solve these problems, new cancer treatments are being investigated by examining molecular targets associated with oncogenes, tumor suppressors, and RNAi. These treatments have many purposes, including inhibiting kinases that promote cell proliferation, strengthening the immune system against cancer, using specific drugs, targeting the delivery of drugs to cancer cells, and reducing the side effects of vaccine disease. Additionally, many mechanisms contribute to chemoresistance, such as drug inactivation, multidrug resistance, inhibition of apoptosis, altered drug metabolism, epigenetics and drug targeting, enhanced DNA repair, and gene amplification [126].

There are many ways to detect and monitor the development of early-stage cancer. By using this technique effectively, the development of chemoresistance can be prevented or its development can be controlled in patients who have already developed it. Among those techniques are dysfunction, interaction of drug effects, changes in drug targets, drug efflux, DNA damage repair, inhibition of cell death, epithelial to mesenchymal transition, and intrinsic tumor cell heterogeneity in drug resistance. Additionally, the modification of epigenetic factors that may cause drug resistance and their ability to contribute to the emergence of cancer progenitor cells unaffected by cancer therapy have been described [125,126]. Drug resistance occurs when a disease becomes resistant to drug treatment. This phenomenon has been observed not only in cancer but also in other diseases. Resistance was first introduced when bacteria showed resistance to specific antibiotics. Drug efflux has been found in microbial and human cancer cells and is a defense mechanism distinct from other mechanisms specific to certain diseases [120]. Although chemotherapy is initially effective against many types of cancer, resistance can occur due to several mechanisms, including DNA mutations and metabolic mutations that promote inhibition and degradation of the drug. Drug resistance occurs through drug inactivation, drug target modification, drug efflux, DNA damage repair, cell death inhibition, and the transition of the cancer cell from the epithelial state to the EMT. Developing anti-cancer drugs and current methods to solve the problems is an essential research area [120,121,122,123,124,125].

The role of cellular heterogeneity in cancer cells in the emergence of drug resistance has also been considered. Finally, the impact of epigenetics on cancer resistance and its essential role in the survival of cancer progenitor cells that are not affected by cancer treatments are important parameters too [120]. Another significant issue is understanding the complex mechanisms of chemoresistance for almost all drugs used to treat the most lethal cancers to prevent or maintain them [123]. It is essential to understand that chemoresistance is present from the early stages of chemoresistance and, therefore, requires some methods to detect it. The main methods used for the early detection of anti-cancer drugs include new cancer cell screening, cancer biomarkers, positron emission tomography (PET), and high throughput pharmacogenomic CRISPR screening [123] (Table 1).

(a)Fresh Tumor Cell Culture Assay (Tumor Chemosensitivity Assay): New tumor culture screening technology has been widely used for decades and good results have been achieved. However, its limitation is that it cannot predict the side effects of the drugs given to patients. Many randomized clinical trials and omics technologies, such as pharmacogenetics, have been proposed to solve this problem. This technology will be adapted to each patient’s needs and drug combination, providing a more in-depth understanding of the interaction between the patient’s genome and the drug used [127]. More than 50% of cancers are resistant to chemotherapy before chemotherapy is initiated. In additional cases, this resistance (so-called secondary resistance) develops after initiating treatment [120,127]. To obtain a new tumor, an oncologist must conduct a blood test, which requires proper planning for transferring the samples for a quick check. This method aims to obtain cancer cells from different tumor types that preserve their physiological properties [128]. Preparation methods will differ depending on the nature of the cancer cells; However, simple steps, such as cell extraction, incubation with antibodies, and cell viability assessment control, remain the same in all cell types. Many antibiotics are used because the primary purpose is to build the immune system. It is valuable to add that the gels used to treat the disease were also used in this experiment because the aim is to determine the anti-cancer effect. In each method, in addition to measuring cell viability, the molecular structure of tumor cells is also analyzed to indicate the growth or death of the cells and the hand activity level is also determined [127].Among the various pathways, thymidine incorporation into cellular DNA and the depletion of cellular ATP are the most commonly used mechanisms. The presence of protection can be confirmed by incorporating thymidine into cellular DNA or by the absence of a decrease in cellular ATP levels. The culture of new tumor cells is suitable for many types of cancer and, given their role in the cellular response, their predictive value can be measured as a precise measure of allergic reactions [128,129,130]. The advantage of this method is that it can be used not only in tumors (such as ovarian cancer, etc.) but also in hematological malignancies [130].(b)Cancer Biomarker Test: Biomarkers, such as DNA, RNA, peptides, genes, and proteins, can clearly understand a person’s cancer and its specific type. It is essential to understand this information because each person has a unique genome. Therefore, cancer treatment can be personalized according to the patient. This approach recognizes that chemotherapy resistance may vary from patient to patient, depending on the unique genome. Therefore, although the principle of cancer treatment remains the same, treatment details may differ due to genetic differences between people. These specific biomarkers provide essential information that can help physicians choose appropriate treatments, including using specific medications for cancer patients [129]. Cancer biomarkers also work as clinical tools that can measure the stage of cancer (blood in tissue) and predict, for example, a patient’s risk of developing cancer. They can also measure the resistance of cancer cells in the patient’s treatment. By following this approach, appropriate treatments can be selected for each specific cancer patient. This approach, with the help of omics technology, allows a better understanding of the needs of cancer patients and the use of different types of treatments. Thus, this approach may help increase the effectiveness of treatment and extend the patient’s life [6]. Two main groups of biomarkers are used to treat cancer patients: (1) anti-cancer biomarkers that help detect and treat cancer, in addition to diagnosing cancer and predicting the patient’s response to medications and (2) pharmacokinetic biomarkers that can help determine the optimal dose for cancer treatment. The biggest challenge facing these two biomarkers is that they are less helpful when applied to cancer than leukemia patients. This difference can be attributed to the occurrence of different types of cancer [128,129,130].In leukemia, many cancer cells can be easily found in the peripheral blood; thus, the use of anti-biomarkers is easier. In contrast, detecting these cells in the peripheral blood of cancer cells is more difficult because they can only follow the later stages of the disease. In this case, the only option is to have a biopsy or, in some cases, remove the tumor. However, analysis of these cells to determine the appropriate treatment often does not help diagnosis due to delays in diagnosis. Overall, each prognostic biomarker has advantages and disadvantages and is helpful for a particular stage. For example, genetic signatures are not valuable for cancer (due to difficulties in obtaining tissue) and do not serve as predictive disease biomarkers. On the other hand, tumor DNA genotyping appears to be more reliable as a predictive biomarker in these patients. Further research, especially in the field of predictive biomarkers, may help select the most appropriate treatment for cancer patients [130,131].(c)Positron Tomography: PET plays an essential role in the treatment strategy of cancer patients, especially in cancer treatment. This critical step allows clinicians to make informed results about the most appropriate treatment for individual patients. PET can help physicians make an accurate diagnosis of cancer by improving early detection or determining the stage of the disease. Therefore, PET scans may help select curative treatments for early-stage tumors or palliative approaches for invasive disease. In addition, since oncology treatment is complex and challenging, early diagnosis is essential in increasing the effectiveness of treatment and reducing financial costs for patients [6,129,130]. PET/CT imaging technology is beneficial in the early diagnosis of cancer. This method is based on the observation that cancer cells will absorb more radiation, resulting in a brighter image. This brightness can help identify cancer cells in the early stages of the disease. In addition, physicians can offer appropriate treatment to patients with cancer. This helps choose the proper treatment and reduces the risk of using anti-cancer drugs by avoiding inappropriate medication or dosage. Remember, although histopathology provides a reliable assessment of cancer treatment, only a smaller number of patients (20–40%) achieve a complete pathological response. Therefore, increasing accessibility to early diagnosis and treatment can improve the quality of treatment. PET/CT imaging is one of the methods that can help achieve this goal [130].(d)High Throughput Pharmacogenomic CRISPR Analysis: High throughput CRISPR technology is a promising new genomic approach for cancer research, especially in the summary of hematological malignancies. This technology can potentially be used in many types of cancer for fundamental purposes, such as identifying regulatory genes that can serve as biomarkers for malignant transformation and developing therapeutic targets and new drugs. It is an essential tool in biological research, especially in the biotechnology and pharmaceutical industries. It has also become routinely used to examine hematological cancers in recent years, making early cancer detection one of its main applications. CRISPR/Cas9 currently provides many genome editors; these include the CRISPR/Cas9 nucleotide sequence editor, CRISPR/Cas base editor (BE), CRISPR primer editor (PE), and CRISPR interference (CRISPRi) (such as CRISPRa, CRISPRa, and CRISPRr). They are also used in many biological sciences, such as early cancer detection, cancer diagnosis, and the development of new drugs to treat hemorrhagic cancer [118].

### 4.2. Rational Drug Design and Personalized Medicine Approach

It is the body of Rational Drug Design and is used to treat, prevent, or diagnose diseases. This goal is usually achieved by using drugs derived from natural or synthetic materials. Drug development principles revolve around making it safe, non-toxic, non-dangerous, or reducing the incidence of side effects. In addition, the drug must be chemically and metabolically stable to prevent the formation of harmful substances in the body. Depending on the specific drug, it must be soluble in water or lipids because it does not release into the blood or, thus, facilitates penetration into cell membranes [6,130,131,132].

Additionally, drugs must be able to target specific drug molecules and, then, be distributed throughout the body [133]. When drugs enter the body, they can cause two different reactions. The first, called pharmacodynamics, refers to the effect of drugs on the human body, including the specificity of their mechanisms, the relationship between doses, and the resulting side effects. The second type is pharmacokinetics, which focuses on the interaction between the human body and drugs, including the absorption, distribution, metabolism, and elimination of drugs after consumption. The latter is often called the ADME process, which is specific to each drug and depends on its chemical structure and base material. This proc1ess helps elucidate each drug’s pharmacokinetics and safe function in the body [133,134]. The goal of drug development is to create new treatments. A combination of chemistry, biology, bioinformatics, mathematical biology, experiments, translation, and clinical models will eventually be needed to achieve this goal. However, in recent years, drug development has relied on new technologies, such as computer modeling and bioinformatics. These two disciplines reduce the cost and time required for drug development. Despite significant scientific advances, especially in biotechnology, drug discovery is still expensive and time-consuming. Therefore, developing new drugs requires a significant amount of time, as well as its financial impact. Therefore, the probability of failure in developing new drugs is still high due to the abovementioned reasons [135]. An essential part of drug development revolves around identifying biological targets. The design process requires the construction of molecules with the appropriate conformation and charge that will then interact with and bind to the biological target [133].

Personalized medicine is a new approach to patient care that uses a person’s unique characteristics, including genetics, to inform clinical decisions, aiming to deliver the proper treatment to the patient at the right time. Medicine is currently growing and expanding with significant resources for its advancement. One of its main products includes diagnostic research and diagnostic and predictive biomarkers. Personalized medicine can be used for many types of treatments and has become standard practice in treating many conditions, including gastrointestinal diseases. Its importance is especially true in the field of oncology, where it has an impact on early diagnosis and prevention. This importance stems from the understanding that the selection of appropriate surgical and chemotherapy strategies is important in ensuring short-term reduction and long-term results. Selecting appropriate patients for treatment in order to maximize efficacy and minimize toxicity has long been recognized as an important aspect of patient management, especially this one. It is important to identify patients who can benefit from this approach in personalized medicine [136,137]. Personalized cancer care represents the best in medicine because it is the most advanced treatment based on the concept that every cancer patient is different. Over the past few years, intense research by cancer scientists has uncovered a wide array of molecular and cellular mechanisms of cancer, including tumorigenesis, cell proliferation, and metastasis. This process revolves around many factors such as genetic mutations, chromosomal abnormalities, epigenetic changes, and interactions between tumors and hosts [136,137].

But, until recently, clinicians had limited resources to determine which patients would benefit from chemotherapy and which patients would be harmed by anti-cancer drug use. There have been exciting advances in personalized cancer treatment, such as diagnostic and predictive biomarkers. These biomarkers allow treatment to target patients who will benefit most. As a result, survival rates improved, and the practice became part of routine medical practice. Personalized cancer medicine has great potential, especially in the prevention and treatment of cancer, and will undoubtedly have an impact on future treatments [136]. In personalized medicine, changes in genes or proteins in cancer patients can be considered important in the application of what we call personalized cancer treatment. This approach relies on the variables described above to determine the most appropriate treatment for each patient [134,137]. The basic approach in personalized medicine is the use of historical data for the diagnosis and treatment of patients. The basic steps of personalized medicine consist of (1) obtaining personalized medical information (such as genome sequencing technology, transcriptomics, proteomics, and metabolomics technology), (2) collecting physiological and lifestyle information, (3) storing the obtained data, (4) creating links to omics data between treatments, and (5) conducting omics tests and diagnostics [138]. Finally, personalized medicine is developing rapidly and gaining importance in the treatment of diseases, especially cancer. Early diagnosis of breast cancer can go a long way toward implementing appropriate and effective treatment strategies. Therefore, this new treatment, along with early diagnosis, will lead to further advances in the prevention and treatment of many cancers [134,135,136,137].

### 4.3. The role of Predictive Biomarkers in Guiding Medical Decisions

Biomarkers work as measurable indicators, measured accurately, safely, and non-invasively. Biomarkers enable the collection of sufficient information about the patient by measuring biological characteristics for clinical use. This test helps doctors obtain an accurate and reliable picture of a person’s health before taking further action, especially invasive procedures, such as biopsies and surgery. There are six main groups of biomarkers used for biological purposes: susceptibility/risk, diagnostic, prognostic, monitoring, pharmacodynamic, and predictive biomarkers [123]. Since this review focuses only on the last category of biomarkers (predictive biomarkers), to discuss the first five types is beyond the scope of this review so has not been discussed. The purpose of making predictive biomarkers is to know the effect of the intervention or its effect on the patient’s drug or medical product. These markers play an important role in making treatment decisions, allowing clinicians to make the right choice for patients. The basis for predicting markers is advances in genomics and proteomics, which can identify genes and proteins associated with different stages of cancer. Advantages of these biomarkers include the ability to discriminate between benign and malignant, and metastatic and non-metastatic tumors. They can also be detected in blood vessels due to the size of the tumor. The importance of predictive biomarkers is in assessing the tumor’s ability to respond to drugs. This is a form of self-care in nursing. On the other hand, a limited number of biomarkers are available [138]. However, due to the variability of patients’ response to chemotherapy, predictive biomarkers are urgently needed to predict drug response. This is because a new generation of cancer drugs are effective in only a small percentage of patients. This fact is important because there are adverse reactions associated with drugs in this class; when given to unqualified patients, they not only fail to achieve therapeutic results but may also worsen the condition of these patients [138,139]. For example, HER2/neu status in breast cancer is determined by overexpression of the HER2/neu protein in some types of breast cancer. Therefore, analysis of the protein can be used to determine the effectiveness of a treatment such as trastuzumab (Herceptin). Patients diagnosed with HER2/neu-positive breast cancer may benefit from early detection and subsequent administration of trastuzumab, which has the potential to improve clinical outcomes [140]. Another example is mutations in the EGFR gene in non-small-cell lung cancer may serve as a predictive biomarker for treatment with drugs such as erlotinib (Tarceva) and gefitinib (Iressa). Patients with EGFR mutations may respond better to these drugs, leading to faster clinical outcomes [140].

### 4.4. Lifestyle Changes and Medical Support Improve Treatment Results

Cancer diagnosis and treatment not only affect the patient’s physical and mental activities but also bring with them side effects that cause serious limitations. These restrictions may lead to the temporary suspension or even discontinuation of the drug, which can cause serious harm to the patient’s health. Many studies show the benefits of physical activity, exercise, and exercise therapy after treatment for people treated for cancer in both acute and chronic stages. These plans are not only feasible but also recommended. Additionally, diet plays an important role in all stages of cancer treatment. By combining proper nutrition with physical activity in the form of physical therapy and exercise, the negative effects of treatment can be prevented and reduced [141].

To obtain specific advice about a particular type of cancer or its side effects, it may be helpful to seek personal advice from an expert on nutrition and exercise strength. The ACS’s Nutrition and Physical Activity Guidelines for Cancer Survivors recommend a healthy lifestyle that includes maintaining a healthy weight, partaking in regular exercise, and following a diet rich in vegetables, fruits, and whole grains [141].

## 5. Clinical Importance and Future Direction

### 5.1. Clinical Research and Treatment Results of Metastatic Colorectal Cancer Cells

In addition to the anti-cancer drugs mentioned above, cancer cells can also exhibit resistance to 5-fluorouracil (5-FU). In addition, 5-FU is a synthetic fluorinated pyrimidine analog that works by inhibiting DNA replication, thereby inserting fluorinated nucleotides instead of thymidine into the DNA structure, causing cell death. Therefore, it is not surprising that there is a relationship between TS expression and 5-FU resistance. TS, as a judicial term, plays an important role in this. Patients with lower TS expression had improved overall survival (OS) in response to 5-FU treatment compared with patients with higher TS expression in the tumor. While monoclonal antibodies represent a group of therapeutic targets, small molecule inhibitors are good treatment options after treatment. Importantly, the appearance of monoclonal antibodies can inhibit VEGF and EGFR. As a result, the OS of colorectal cancer (CRC) patients was extended to three years thanks to this therapeutic intervention. Additionally, the side effects of the treatment were reduced compared to the side effects caused by chemotherapy. One example of such treatment is bevacizumab, a VEGF-specific anti-angiogenic drug that inhibits tumor growth rate. Additionally, Van der Jeught and colleagues evaluated the duration of response and improvement in OS in CRC patients by evaluating data from three clinical trials [142]. These studies evaluated patients receiving fluorouracil/leucovorin alone or in combination with bevacizumab. Many new treatments have emerged in recent years due to the low survival rate of monotherapy. Current treatments for prostate cancer use androgen receptor signaling and combination therapy with two or three ADT drugs. Adherence to established agents, such as antibiotics and ARSI (androgen receptor signaling inhibitor), improved overall survival. Additionally, many new treatment strategies are now available for patients with metastatic castration-resistant prostate cancer. In this case, the traditional approach is limited to chemotherapy only. However, recent advances in new treatments have increased the survival rate after chemotherapy. These options include a variety of treatments, including radioligand therapy with ARSI, PARP inhibitors, and Lu-PSMA. Research on the role of immunity in prostate cancer is ongoing; however, the use of bispecific T-cell engagers (BiTEs) is new in this field of expertise. Prostate cancer treatment is known for its complexity and potential success, mainly due to the many new treatments available. Many new treatment strategies are currently being investigated in the field but have not yet become standard in clinical practice. Ongoing research is focused on further examining the field of prostate cancer immunotherapy [142]. In breast cancer, despite advances in the diagnosis and treatment of the disease, a significant number of patients cannot be treated, which causes the disease to spread and recur, reducing the chance of survival. Classification of breast cancer, according to its physiological significance, particularly the presence or absence of classic signs, has historically been very popular. Immunohistochemical markers, such as estrogen receptor (ER), progesterone receptor (PR), and HER2, play an important role in this classification process. However, it is widely accepted that cancer stem cells (CSCs) play an important role in the regeneration of cancer cells. Breast CSCs (BCSCs) represent a small subset of cells that contain stem cells in breast tumors, characterized by the CD44^+^/CD24 expression profile. It has been demonstrated that the presence of BCSCs in the tumor environment can enhance chemotherapy resistance [143]. Additionally, the presence of high ALDH activity (ALDH + phenotype) leads to resistance to chemotherapy, hormonal therapy, and radiation, allowing tumor regrowth after initial treatment and, thus, reducing the number of injured hands. This phenomenon can lead to relapse [144].

Ovarian Cancer: In ovarian cancer, tumor-associated macrophages (TAMs) constitute most of the immune cells found in the ovarian microenvironment. These cells have a high degree of plasticity and can adopt an immune phenotype similar to M2 macrophages when stimulated by colony-stimulating factor 1 released by tumor cells. The role that these cells play in ovarian cancer and chemotherapy is also very important. M2-like TAMs often exert tumor-promoting activities by releasing various cytokines, chemokines, enzymes, and exosomes. Cytokines, chemokines, enzymes, and exosomes directly interact with microRNA to activate ovarian cancer [144].

M2-like TAMs are important for ovarian cancer metastasis in the peritoneum as they promote cancer cell spheroid formation and attachment to the omentum, the site of metastatic spread. Additionally, TAMs interact with other immune cells, such as dendritic cells, natural killer cells, and lymphocytes, reducing uptake and causing an immune response. Various studies have shown that TAMs play a positive role in ovarian tumors, with a link between high TAM levels in tumors and poor prognosis [145]. TAMs, myeloid-derived suppressor cells (MDSCs), regulatory T-cells (Tregs), stromal cells, cancer-associated fibroblasts (CAFs), and endothelial cells comprise the TME [146,147]. Cancer development, therapy resistance, and immunoescape are all aided by these cells [148,149]. CSCs have an important role in tumor initiation and progression, as well as immune evasion and resistance to chemotherapy and radiation [150,151]. CSCs are a subgroup of tumor cells that cause tumor resistance and recurrence. A cancer cell with the stem cell characteristic (including its similarities with long-lived postmitotic cells) may divide and create a wide range of progenies; it is responsible for disease progression, tumor resistance to therapy and the immune system, and disease recurrence [152].

### 5.2. Challenges and Opportunities in Interpreting Clinical Trials

Major challenges in interpreting new evidence often arise from individual biases rather than institutional constraints. Personal problems often arise from the inability to conduct, organize, use, and evaluate research data. On the other hand, organizational problems arise from limited access to research evidence and inadequate resources. To overcome these problems, cooperation must be ensured between policymakers and practitioners at all levels and stages of the research process. The findings of this study highlight the importance of identifying problems and opportunities to prioritize the use of findings. Improving the translation of research findings into clinical practice requires effective collaboration and cooperation among stakeholders [153].

### 5.3. Potential Future Directions and New Therapeutic Approaches

(a)The development of immunotherapy represents a promising future [154]. The combination of chemotherapy and radiation with immunotherapy is one of these methods. This approach is to reduce tumor cells, causing them to die while increasing the glucose level that natural killer cells need to kill cancer cells. Additionally, other methods include administering nutrients that can inhibit the glycolytic process of the immune system [155]. Epigenetic therapy also has the potential to find effective solutions to cancer [156]. Determining the strategy to improve the effect of epigenetic factors in these tumors is a good example in this field [154];(b)Regarding epigenetic therapies, the traditional approach they take is to promote the expression of inhibited tumor cells and restore their right to grow [157]. An example of this is the removal of DNA methylation, which leads to reduced transcription of the gene [158];(c)Activity, especially exercise, has been shown to be beneficial to cancer patients. Exercising before surgery can increase the body’s strength, resulting in an overall improvement in the patient’s health before and after surgery [159]. There is also good evidence to support the use of exercise as a way to prevent cancer. Epidemiological studies have shown that exercise is effective in controlling symptoms and improving the quality of life in cancer patients, especially prostate cancer patients [160];(d)The emergence of multi-omics, a set of diagnostic tools that include genomic, epigenomic, transcriptomic, epitranscriptomic, and proteomic networks, has revolutionized cancer treatment. This technology has made it possible to diagnose and treat diseases such as cancer [161].

## 6. Conclusions

Good evidence shows that every cancer patient is unique and an individual. Therefore, using omics technologies to deliver personalized cancer treatment to each patient is a promising and effective way to prevent and overcome this disease. To achieve this goal, further research is encouraged, especially in the field of omics and personalized cancer medicine. Finally, early diagnosis and treatment of cancer at the earliest stage using modern science has increased the life expectancy of this special group of patients. Additionally, the use of personalized medicine opens up new possibilities to increase the success of treatment for these patients.

## Figures and Tables

**Table 1 biomedicines-12-00183-t001:** Main methods used for the early detection of anti-cancer drugs.

Number	Type	Mode of Action	Advantages	Limitation	References
1	Fresh Tumor Cell Culture Assay (Tumor Chemosensitivity Assay)	To collect cancer cells from fresh cell types that preserve their physiological properties.	Good results over the last decades. Simple steps for all cancer cell types. Can be used for all cancer cell types.	Lack of predicting the drug side effects given to the patients. Preparation method steps vary depending on different cancer cell types.	[120,127,128]
2	Cancer Biomarker Tests	To measure biomarkers, such as DNA, RNA, peptides, genes, and proteins.	Clinical biomarkers for predicting cancer stage (e.g., blood in tissue) and to predict a patient’s risk of cancer development. To measure the chemoresistance of cancer cells to drugs. Combining the first two above approaches with the assistance of omics technologies can increase each patient’s life survival.	Detecting cancer cell types is not easy in the peripheral blood of cancer cells, unless with biopsy or removal of the tumor; therefore, just the opposite of leukemia, the other cancers are detected in late stages.	[6,129]
3	Positron Emission Tomography (PET)	This method is based on cancer cells absorbing more radiation, resulting in a brighter image. This leads to more accurate, reliable, and early detection of cancer in patients.	To help clinicians to make an accurate diagnosis. To determine the stage of cancer. To help choose the most appropriate curative therapy for early-stage tumors. To help palliative methods for invasive disease. To reduce the cost of therapy by choosing the most accurate therapy method.	Higher costs of this diagnostics method.	[6,129,130]
4	High Throughput Pharmacogenomic CRISPER Analysis	Novel genomic method for cancer research.	To detect regulatory genes that can act as biomarkers for malignant transformation. Developing therapeutic targets of new drugs. Early detection of hematological cancers.	Higher costs of this novel research method.	[118]

## Data Availability

Not applicable.

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
