# Peer review of "Overcoming Chemotherapy Resistance in Metastatic Cancer: A Comprehensive Review"

_biomedicines, 2024, doi:10.3390/biomedicines12010183_

Round 1

Reviewer 1 Report (Previous Reviewer 1)

Comments and Suggestions for Authors

This manuscript is suitable for publication in Biomedicines after minor revision.

 1) lines 102 and 375) Which is right, epidermal growth factor receptor (EGDR) or epidermal growth factor receptors (EGFR)?

2) line 345) What are passive and active tumor-targeting skills? Please explain them in more detail.

3) line 357) What is the PEGlyation?

4) The authors should present a summary figure or table to help the understanding of readers for your manuscript.

Author Response

Reviewer 2 Report (Previous Reviewer 2)

Comments and Suggestions for Authors

-Please review this sentence:  Down-regulated expression of CD95 has been observed in leukemia neuroblastoma cells, lane-204

-Please review this sentence:  of action due to their  passive and active tumor-targeting skills, which can reduce,  lane-345

-Please review this sentence: Some of the best 484 known receptors associated with this mechanism , lane 374

-This paragraph could be better structured :

“Beta-blockers, which target beta-adrenoreceptors, are widely used against cardiac  conditions. These drugs can induce anti-proliferative and anti-migratory effects and in- crease survival in experimental animal models of pancreas and colorectal cancer (79). As- pirin is a drug used in cardiovascular disease to treat and prevent myocardial infarctions,  targeting COX enzymes 1 and 2. COX1 produces thromboxane A2 in platelets, leading to  platelet aggregation and adherence to tumor cells (80, 81). COX2 is responsible for pro- ducing prostaglandin E2, which promotes proliferation. Aspirin has been studied for its  therapeutic potential in established or chemically induced tumors, and post-diagnosis use  is associated with favorable survival in patients with colorectal cancer. However, most  studies are retrospective and heterogeneous, with conflicting findings (84). the role of as- pirin as a therapeutic cancer agent was shown previously (82, 83).  Beta-blockers target beta-adrenoreceptors, which are activated by catecholamines and promote tumorigenesis. Blocking beta-adrenoreceptors can induce anti-proliferative and anti-migratory effects and increase survival in experimental animal models of pan- creas and colorectal cancer (84, 85).”

-Some concepts are repeated in this paragraph: 606-610 “Ibuprofen, a non-selective COX

inhibitor, has been shown to inhibit prostate cancer cell growth and exhibit anti-tumor effects through anti-angiogenesis, apoptosis, and reduced cell proliferation. Ibuprofen, a non-selective COX2 inhibitor, has been shown to have antineoplastic properties in various cancer types, including fibrosarcoma, hepatoma, colon, ovarian, and pancreatic cancer  (114).”

Author Response

This manuscript is a resubmission of an earlier submission. The following is a list of the peer review reports and author responses from that submission.

Round 1

Reviewer 1 Report

Comments and Suggestions for Authors

1) This manuscript is well-written and contains some useful information. However, it is the fact that this manuscript is lacking in the depth and the novelty. For example, in the 3. DRUG EFFLUX PUMP, the authors should write the signaling pathways associated with MDR in more detail such as Akt, NF-κB, etc. To be honest, I failed to find the novelty in all sections, so I suggest that the authors should rewrite this manuscript.

2) It is not clear that the signaling pathways selected by the authors are suitable for targeting metastatic cancer cells or for overcoming and preventing chemotherapy resistance. For example, the cell death pathways, the oncogenic signaling pathways, exosomes, etc. Most of them are involved in non-metastatic cancer cells.

3) The authors should carefully check the references, for instance, see Refs 85, 177, 181, etc.

Reviewer 2 Report

Comments and Suggestions for Authors

The review addresses different mechanisms involved in the acquisition of resistance to different chemotherapeutic treatments.

This is a topic of great interest since the acquisition of chemoresistance is one of the most important mechanisms for therapeutic failure in cancer and the development of metastasis.

The review is complete in addressing the main mechanisms involved in the acquisition of resistance to chemotherapy; however, some of the sections are poorly represented and could be improved.

GENERAL COMMENTS

In the DRUG EFLUX PUMP chapter, the authors focus only on the MDR gene, but there are many others, such as the MRP subfamily of proteins that are highly upregulated in chemoresistance in many tumors, that deserve further mention.

CELL DEATH PATHWAYS

Dysregulation of cell death is one of the best-studied mechanisms of resistance to chemotherapy. This area is underrepresented in this review. For example, one of the most significant alterations of cell death by apoptosis is represented by alterations of the Bcl2 family proteins, and only Bcl2 is mentioned in this chapter, and in a very superficial way. Perhaps it would be appropriate to expand the chapter on Bcl2 family alterations as a mechanism of resistance to chemotherapy in both solid and hematologic tumors.

Moreover, the chapter on autophagy is underrepresented, since it is known that this mechanism, i.e. the blockage of autophagic flux, represents a new pathway of resistance to chemotherapy with important references in recent years.

The chapter on DNA repair should also be improved, as it mixes the concepts of homologous and non-homologous recombination pathways with other minor repair pathways without much clarity, and the references cited in this section are not the most representative.

In the section on CSCs, reference 177 is incomplete and there are many other references more representative than the one given to illustrate the influence of CSCs on the EMT phenomenon in the acquisition of chemoresistance.